# Honeybee gut bacterial strain improved survival and gut microbiota homeostasis in *Apis mellifera* exposed *in vivo* to clothianidin

Sarah El Khoury,[1,2] Jeff Gauthier,[1,2] Pierre Luc Mercier,[1,2] Stéphane Moïse,[3] Pierre Giovenazzo,[2] Nicolas Derome[1,2]

**ABSTRACT**    Pesticides are causing honeybee mortality worldwide. Research carried out on honeybees indicates that application of pesticides has a significant impact on the core gut community, which ultimately leads to an increase in the growth of harmful pathogens. Disturbances caused by pesticides also affect the way bacterial members interact, which results in gut microbial dysbiosis. Administration of beneficial microbes has been previously demonstrated to be effective in treating or preventing disease in honeybees. The objective of this study was to measure under *in vivo* conditions the ability of two bacterial strains (the *Enterobacter* sp. and *Pantoea* sp.) isolated from honeybee gut to improve survival and mitigate gut microbiota dysbiosis in honeybees exposed to a sublethal clothianidin dose (0.1 ppb). Both gut bacterial strains were selected for their ability to degrade clothianidin *in vitro* regardless of their host–microbe interaction characteristics (e.g., beneficial, neutral, or harmful). To this end, we conducted cage trials on 4- to 6-day-old newly emerging honeybees. During microbial administration, we jointly monitored the taxonomic distribution and activity level of bacterial symbionts quantifying 16S rRNA transcripts. First, curative administration of the *Pantoea* sp. strain significantly improved the survival of clothianidin-exposed honeybees compared to sugar control bees (i.e., supplemented with sugar [1:1]). Second, curative administration of the *Enterobacter* sp. strain significantly mitigated the clothianidin-induced dysbiosis observed in the midgut structural network, but without improving survival.

**IMPORTANCE**   The present work suggests that administration of bacterial strains isolated from honeybee gut may promote recovery of gut microbiota homeostasis after prolonged clothianidin exposure, while improving survival. This study highlights that gut bacterial strains hold promise for developing efficient microbial formulations to mitigate environmental pesticide exposure in honeybee colonies.

**KEYWORDS**    honeybee, clothianidin, microbiota, dysbiosis, beneficial microbes

The worldwide agri-food industry heavily depends on pollination services provided by honeybees (*Apis mellifera*) (1). The apparent honeybee health deterioration has raised concerns in recent decades (2, 3). Such as, their exposure to common pesticides reduced pollination efficiency (4, 5), leading partly to global honeybee colony declines (6). Neonicotinoid insecticides are frequently used in various agricultural productions (7, 8), but unfortunately, their long half-lives and extensive usage in the environment pose a worrying threat to honeybees (9).

Accumulated data so far suggest that sublethal neonicotinoid exposure impairs honeybee physiological functions such as digestion (10), behavior (11, 12), cognition (13), neuronal communication (14), and most importantly detoxification (15)

Address correspondence to Nicolas Derome, nicolas.derome@bio.ulaval.ca, or Sarah El Khoury, selkhoury@berkeley.edu.

The authors declare no conflict of interest.

See the funding table on p. 19.

*Please note that the primers used in this work contain Illumina specific sequences protected by intellectual property (Oligonucleotide sequences 2007–2013 Illumina, Inc. All rights reserved. Derivative works created by Illumina customers are authorized for use with Illumina instruments and products only. All other uses are strictly prohibited).*

and immune response (16). Notably, these functions, particularly immune response, are partly regulated by the gut microbiota (17–20), which may play a pivotal role in supporting honeybees under pesticide stress. The honeybee genome possesses enzymes for metabolizing xenobiotics but exhibits lower detoxification gene diversity compared to other insects (21), suggesting that honeybees may depend on factors like microbiota for assistance in breaking down harmful molecules. In recent decades, microbes have demonstrated the ability to degrade chemical compounds in natural environments and have been found in various insect orders (22–25), including within the gut microbiota of the honeybee (26).

The gut microbiota of honeybees is mainly composed of nine phylotypes of bacteria, representing over 98% of the bacterial 16S rRNA gene sequences (27). These bacterial strains are divided into three groups: core members (found in all *A. mellifera*), non-core members (not found in all *A. mellifera*) (28), and low abundance taxa (29). Honeybee gut microbiota is inherited primarily from other honeybees, through nurse feeding, rather than acquired from the external environment. The core members of the honeybee gut microbiota are composed by *Lactobacillus* (Firm-5, Firm-4), *Bifidobacterium* spp., *Gilliamella apicola*, and *Snodgrassella alvi*, whereas *Frischella perrara*, *Bartonella apis,* and *Parasaccharibacter apium* represent the non-core members (20, 28, 30). Latest studies established relationships between the gut microbiota and honeybee health (17, 31–33).

Honeybee gut microbes are fundamental in performing host beneficial functions in digestion (34), nutrient assimilation, detoxification (35), modulating the immune response (17), and preventing colonization of host tissues by pathogens (34). However, these benefits are hindered when the gut microbiota undergoes maladaptive changes in community diversity, i.e., dysbiosis (36). Indeed, previous studies already observed that gut dysbiosis was associated with exposure with neonicotinoids: thiacloprid (37), nitenpyram (38), imidacloprid (39), and clothianidin (33).

Beneficial microbes supplementation was introduced as a promising method that may easily be integrated alongside current agricultural infrastructure and apicultural management practices (40) in order to reduce adverse effects of sublethal pesticide exposure on honeybees (41). By definition, a probiotic is defined as a "live microorganism that, when administered in adequate amounts, confers a health benefit on the host" (42).

In a previous study, we isolated honeybee gut bacterial strains able to completely degrade clothianidin after 72 h under *in vitro* conditions (26). These promising results offer a new perspective on using microorganisms to reduce clothianidin's harmful impact on honeybee colony health. In terms of commercial microbial supplementation, some strains have been reported to cause dysbiosis, resulting in an imbalance between microbiota and host (43, 44). This suggests that certain broad-spectrum microbes may not be suitable for fulfilling the nutritional needs of honeybees (45). Also, researchers have recently started developing genetically modified microbes using CRISPR/Cas9 to degrade pesticides and regulate honeybee pathogen development (46). The selection of promising beneficial microbes to mitigate microbiota dysbiosis and in turn to prevent a disease should prioritize honeybee-derived (i.e., endogenous) strains (47, 48). Among the seven bacterial microbes tested *in vitro* in our previous study (26), two bacterial strains belonging to *Enterobacter* and *Pantoea* genera were selected. Despite the opportunistic characteristic highlighted in honeybee for other strains belonging to the same two genera (28), the two strains tested in this study were selected because both genera have been documented to induce a positive effect on insects' fitness: when (1) isolated from insect microbial gut and (2) used in the insect (fruit fly) diet such as in *Ceratitis capitata* and in *Bactrocera dorsalis* (49–53).

We hypothesized that two strains of *Enterobacter* and *Pantoea* genera isolated from the gut of healthy honeybees, known for their ability to degrade clothianidin *in vitro* (26) , could restore gut microbiota balance and reduce clothianidin toxicity toward honeybees. Additionally, we selected a clothianidin concentration of 0.1 ppb based on our previous findings indicating its significant negative impact on honeybee health (33). Then, considering that (i) up to 30 days post-emergence, adult bees are characterized by

an abundant microbial community (54), and (ii) cage bee average lifespan is about 30 days (55), the present analysis was focused on honeybees sampled after 21 days, with the goal of measuring intensity of chronic pesticide exposure stress rather than cumulative stress, in part due to the limited lifespan of caged bees.

To test our hypothesis, we aimed to investigate to what extent the different parts of the gut bacterial microbiota of bees exposed to clothianidin respond to subsequent bacterial strains supplementation (i.e., curative effect). Monitoring of the gut bacterial microbiota used a metataxonomic approach based on 16S rRNA gene transcripts. This approach aimed to identify functionally active taxa and assess the relative contribution of each bacterial strain to the total activity of the gut bacterial microbiota (33, 56). Co-expression networks based on 16S rRNA gene transcripts were built to detect and quantify changes in community activity. The objective was to evaluate three key aspects: (i) the extent of dysbiosis in honeybee gut microbiota caused by exposure to clothianidin, as documented in El Khoury et al. (33); (ii) the potential restoration of the initial homeostasis state (in the absence of clothianidin exposure) following beneficial microbe administration (indicating a curative effect); and (iii) the possibility of a new transient state characterized by a stable microbial composition resulting from the inoculation of beneficial microbes (also indicating a curative effect), as described in a previous work based on a fish host model (57).

## RESULTS

### Honeybee survival exposed to pesticide and putative beneficial microbes

Honeybees exposed to 0.1 ppb of clothianidin showed significantly higher mortality rate compared to the sugar group (control) ($P < 0.05$; Fig. 1; Table S1). Then, the survival rate of honeybees exposed to clothianidin (0.1 ppb), although supplemented with the *Pantoea* strain (*Pantoea* curative group), showed significantly higher survival compared to the pesticide control group from the T16 until the end of the experiment ($P < 0.01$; Fig. 1; Table S1). Finally, no significant difference was observed for the honeybees exposed to clothianidin (0.1 ppb), although supplemented with the *Enterobacter* strain in comparison with the pesticide control group.

In summary, honeybees exposed to clothianidin (0.1 ppb) experienced higher mortality rates compared to the control group. However, supplementation with the *Pantoea* strain significantly improved survival rates throughout the experiment, whereas supplementation with the *Enterobacter* strain did not show a significant survival benefit over the pesticide control group.

### Clothianidin impact on honeybee gut microbiota after 21 days of exposure

In the midgut, comparisons between the clothianidin pesticide group relative to the sugar control group, interacting amplicon sequence variants (ASVs) (genus level) slightly decreased from 38 to 37 (Fig. 2D and E; Table S2). Exposed midgut ASVs (0.1 ppb) were significantly less connected (DG) relative to the sugar control midgut network (0 ppb) ($P = 0.03$; Fig. 2B, D, and E). Significant positive correlations decreased from 68 to 39, and significant negative correlations slightly increased from 0 to 3 (Fig. 2D and E; Table S3). In the ileum, comparisons between the clothianidin pesticide group relative to the sugar control group, interacting ASVs (genus level) decreased from 66 to 49 (Fig. 3E and D; Table S2). No significant difference was observed for degree (DG) and neighborhood centrality (NC) network parameter (Fig. 3B and C). In the rectum, comparisons between the clothianidin pesticide group relative to the sugar control group, interacting ASVs (genus level) decreased from 38 to 32 (Fig. 4D and E; Table S2). Exposed rectum ASVs (0.1 ppb) were significantly more connected (DG) relative to the sugar control rectum network (0 ppb) ($P = 0.000614$; Fig. 4B, D, and E). Significant positive correlations drastically increased from 38 to 137, and significant negative correlations decreased from 13 to 0 (Fig. 4D and E; Table S3).

Taking these results together, clothianidin exposure for 21 days resulted in reduced interactions among gut microbiota in the midgut, ileum, and rectum of honeybees. In

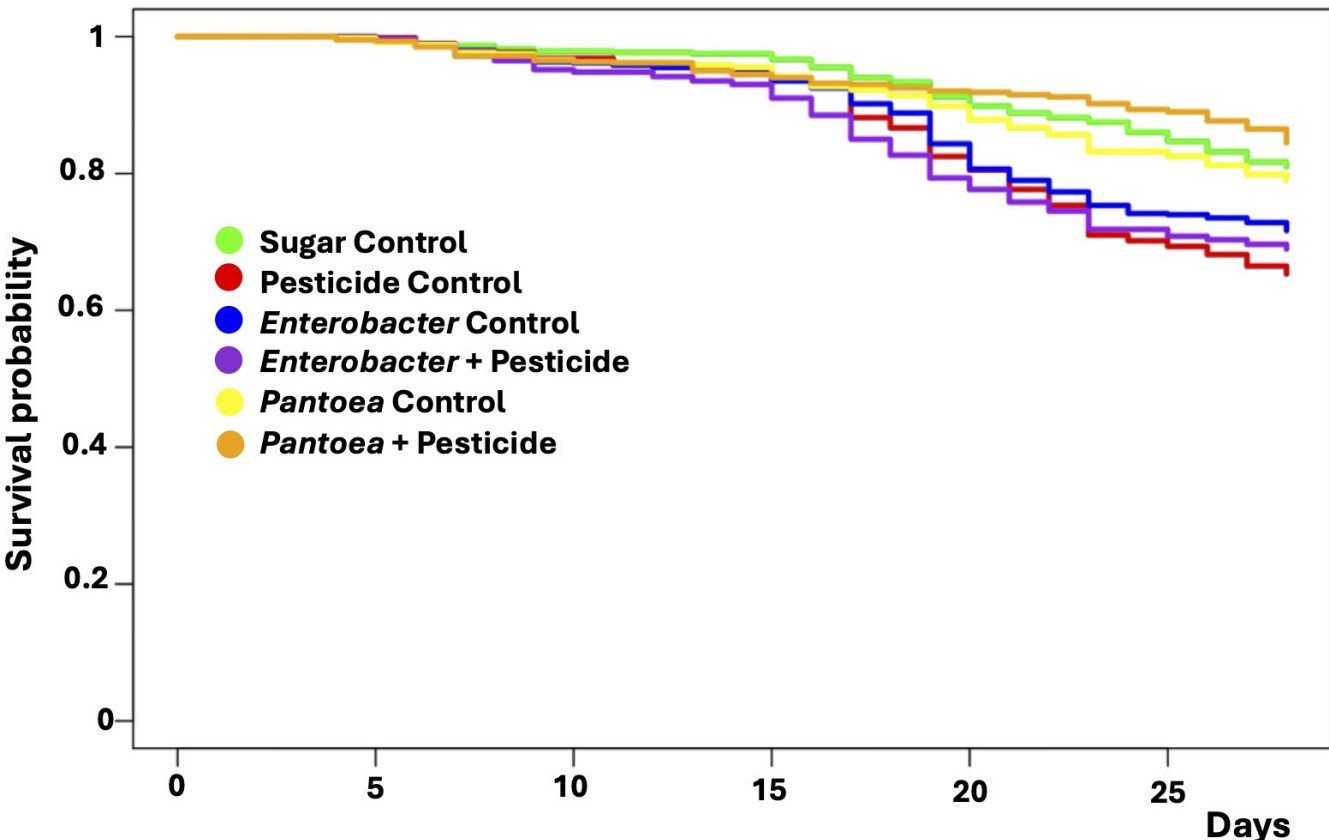

**FIG 1** Kaplan-Meier survival curves of honeybees in each experimental group during the 28-day cage bee experiment. The y-axis represents the Kaplan-Meier estimates of the survival probabilities. The x-axis represents the experimental days. The red, violet, and orange curves represent, respectively, the survival probabilities of honeybees exposed to pesticide only (0.1 ppb), *Enterobacter* + pesticide (0.1 ppb), and *Pantoea* + pesticide (0.1 ppb). The green, blue, and yellow curves represent, respectively, the survival rate of honeybees supplemented with sugar only (1:1) (control), sugar (1:1) + *Enterobacter*, and sugar (1:1) + *Pantoea*. Significant *P* values are represented in Table S1.

the midgut, connectivity decreased significantly, with fewer positive correlations and slightly more negative correlations. In the ileum, interactions between microbial species decreased, whereas no significant network parameter differences were observed. In the rectum, microbial interactions decreased, but exposed rectum microbiota showed increased connectivity, along with changes in correlation patterns.

### Reshaping gut microbiota dynamics: effects of 21-day exposure to putative beneficial microbes showed potential positive effect on gut microbiota homeostasis

First, regarding the *Enterobacter* sp.: in the midgut, comparisons between *Enterobacter* sp. curative group and clothianidin pesticide group showed that interacting ASV number (genus level) decreased from 37 to 35 (Fig. 2D and F; Table S2). Exposed midgut ASVs (0.1 ppb + *Enterobacter*) were significantly more connected (DG) relative to the pesticide control midgut network (0.1 ppb) ($P = 1.58$ e-05; Fig. 2B, D, and F). Significant positive correlations increased from 39 to 144, and significant negative correlations disappeared from 3 to 0 (Fig. 2D and F; Table S3). Regarding network parameters, we observed significant higher values for NC (0.1 ppb + *Enterobacter*) relative to the pesticide control midgut network ($P = 6.31$ e-08; Fig. 2C and H). No significant difference was observed for the closeness centrality (CC) network parameter (Fig. 2A). A general decrease of ASV activity (Fig. 2H) with a significant low increase for *G. apicola*, and a significant decrease for non-core members' ASV activity: *F. perrara*, *Bombella apis* (Fig. 5) were highlighted.

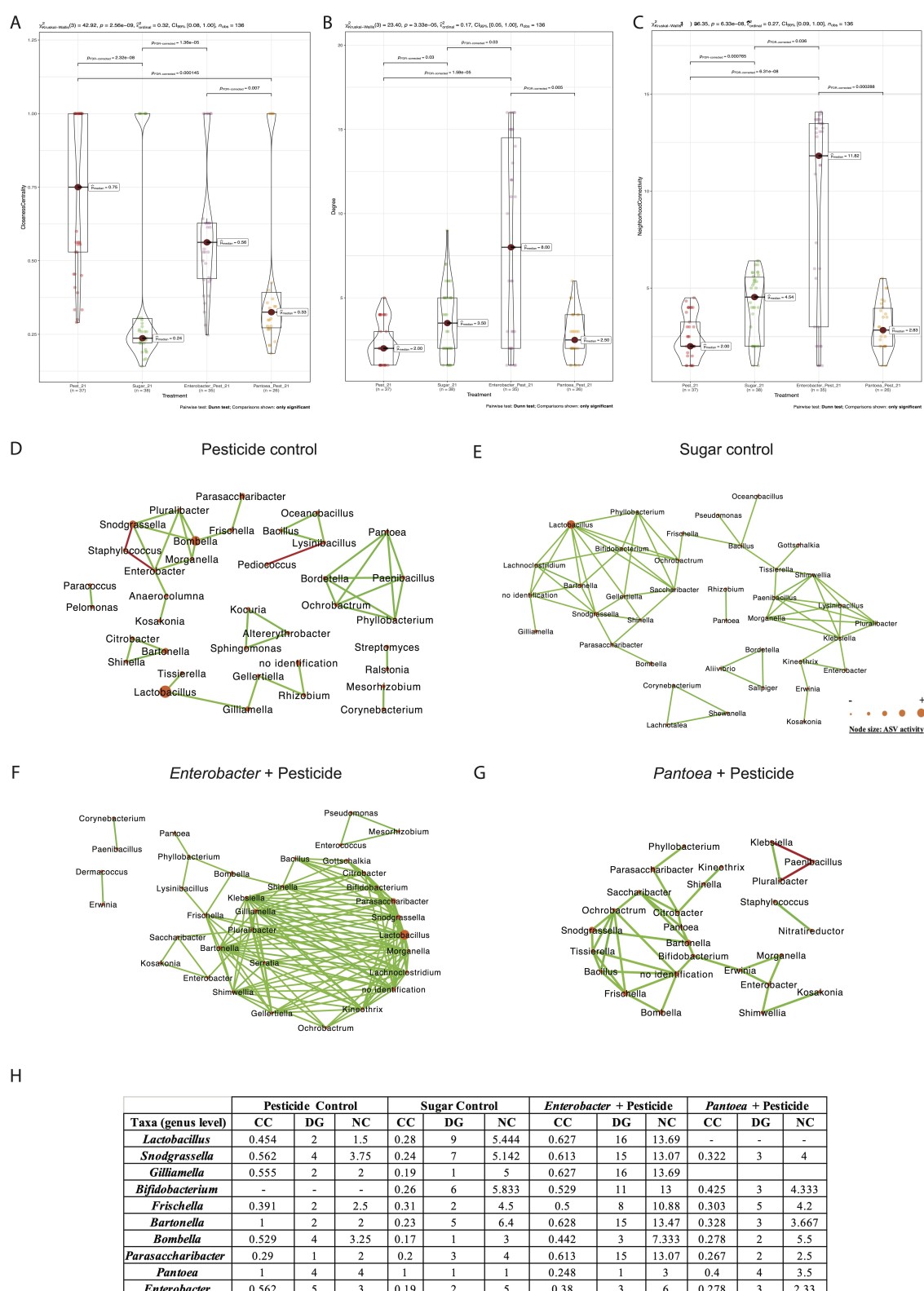

**FIG 2** Violin plot of (A) closeness centrality, (B) degree, and (C) neighborhood connectivity from taxa in midgut microbial networks; microbial networks for the midgut exposed at (D) 0.1 ppb (pesticide control), (E) 0 ppb (sugar control), (F) 0.1 ppb + *Enterobacter* (*Enterobacter* curative group), and (G) 0.1 ppb + *Pantoea* (*Pantoea* curative group) that were generated based on pairwise correlations between the relative abundance of different bacterial genera. We used six replicates (Continued on next page)

**Fig 2 (Continued)**

(five workers per replicate) per experimental condition. Each node represents a bacterial genus. The size of each node is proportional to the bacterial functional activity of each genus. The darker the node, the more interconnected it is. Each edge represents significant positive or negative Spearman correlation coefficients ($-1 \leq r \leq -0.4$) (negative, red) and ($0.4 \leq r \leq 1$) (positive, green); (FDR-adjusted $P$ value < 0.05). (H) Analysis of topological parameter values (CC, DG, NC) of core, non-core members, *Enterobacter*, and *Pantoea* microbe under different experimental conditions (pesticide control, sugar control, *Enterobacter* + pesticide, *Pantoea* + pesticide). Pairwise Dunn test used. Only significant comparisons are shown.

Interestingly, we observed the gain of a core member in interaction in the network: the *Bifidobacterium* sp., which was positively correlated with the *Gilliamella* sp. (another core member), and with five low activity taxa: *Klebsiella, Shinella, Bacillus, Citrobacter*, and an unassigned ASV. All the core and non-core members in correlations increased their number of connections (all positive). *Enterobacter* was negatively correlated with the *Staphylococcus* sp. in the pesticide control midgut. Although the administration of the *Enterobacter* sp. resulted in the inoculation-induced loss of *Staphylococcus* sp. activity within the network, *Enterobacter* exhibited a positive correlation with taxa characterized by low functional activity (Fig. 2D, F, and H).

In the ileum, comparisons between *Enterobacter* sp. curative group and pesticide control group showed that interacting ASV number (genus level) decreased from 49 to 36 (Fig. 3D and F; Table S2). Exposed ileum ASVs (0.1 ppb + *Enterobacter*) were significantly less connected (DG) relative to the pesticide control ileum network (0.1 ppb) ($P =$ 1.35 e-05; Fig. 3B; Table S3). Significant positive correlations decreased from 97 to 30, and significant negative correlations increased from 3 to 7 (Table S3). Regarding neighborhood connectivity (NC), we observed significantly lower values for NC (0.1 ppb + *Enterobacter*) relative to the pesticide control ileum network ($P =$ 5.86e-08; Fig. 3C and H). No significant difference was observed for the CC network parameter (Fig. 3A). In terms of ASV activity, ileum was the most impacted gut region by *Enterobacter* and clothianidin (0.1 ppb) exposure compared to the midgut and rectum (Fig. 5). *Lactobacillus* and *Bartonella* ASV activities have shown to be impacted mainly in the ileum. We observed activity variability (increase and/or decrease) for the following core members: *Gilliamella* and *Lactobacillus* ASVs (*Lactobacillus apis* and *Lactobacillus helsingborgensis*); non-core members: *F. perrara, Bombella, Bartonella*, and *Parasaccharibacter*; and low activity taxa: *Lysinibacillus* and unassigned ASVs (Fig. 5). All core and non-core members were still the most active ASVs within the network (Fig. 3H).

In the rectum, comparisons between *Enterobacter* sp. curative group and pesticide control group showed that interacting ASV number (genus level) was stable, slightly increased from 32 to 33 (Fig. 4D and F; Table S2). Exposed rectum ASVs (0.1 ppb + *Enterobacter*) were significantly less connected (DG) relative to the pesticide control rectum network (0.1 ppb) ($P =$ 0.02; Fig. 4B, D, and F; Table S3). Significant positive correlations decreased from 137 to 50, and four significant negative correlations appeared (Table S3) between low activity taxa. Regarding neighborhood connectivity (NC), we observed significantly lower values for NC (0.1 ppb + *Enterobacter*) relative to the pesticide control rectum network ($P =$ 0.002; Fig. 4C and H). No significant difference was observed for the CC network parameter (Fig. 4A). Rectum was the least impacted gut section in terms of ASV activity. All core and non-core members in correlations in the network were still the most active ASVs (Fig. 4H). However, we observed a significant increase for *Parasaccharibacter* ASV activity (non-core member) (Fig. 4H and 6).

Second, regarding the *Pantoea* sp.: in the midgut, comparisons between *Pantoea* curative group and pesticide control group showed that interacting ASV number (genus level) decreased from 37 to 26 (Fig. 2D and G; Table S2). DG and NC network parameters were not significantly different between (0.1 ppb + *Pantoea* sp.) relative to the pesticide control midgut network (Fig. 2B and C). Regarding closeness centrality (CC), we observed significantly lower values for CC (0.1 ppb + *Pantoea* sp.) relative to the pesticide control midgut network ($P =$ 0.000145; Fig. 2A and H). All core and non-core members were still the most active ASVs in correlations within the network (Fig. 2D, G, and H). Exposure to clothianidin (0.1 ppb) and the *Pantoea* sp. differentially impacted core members' activity,

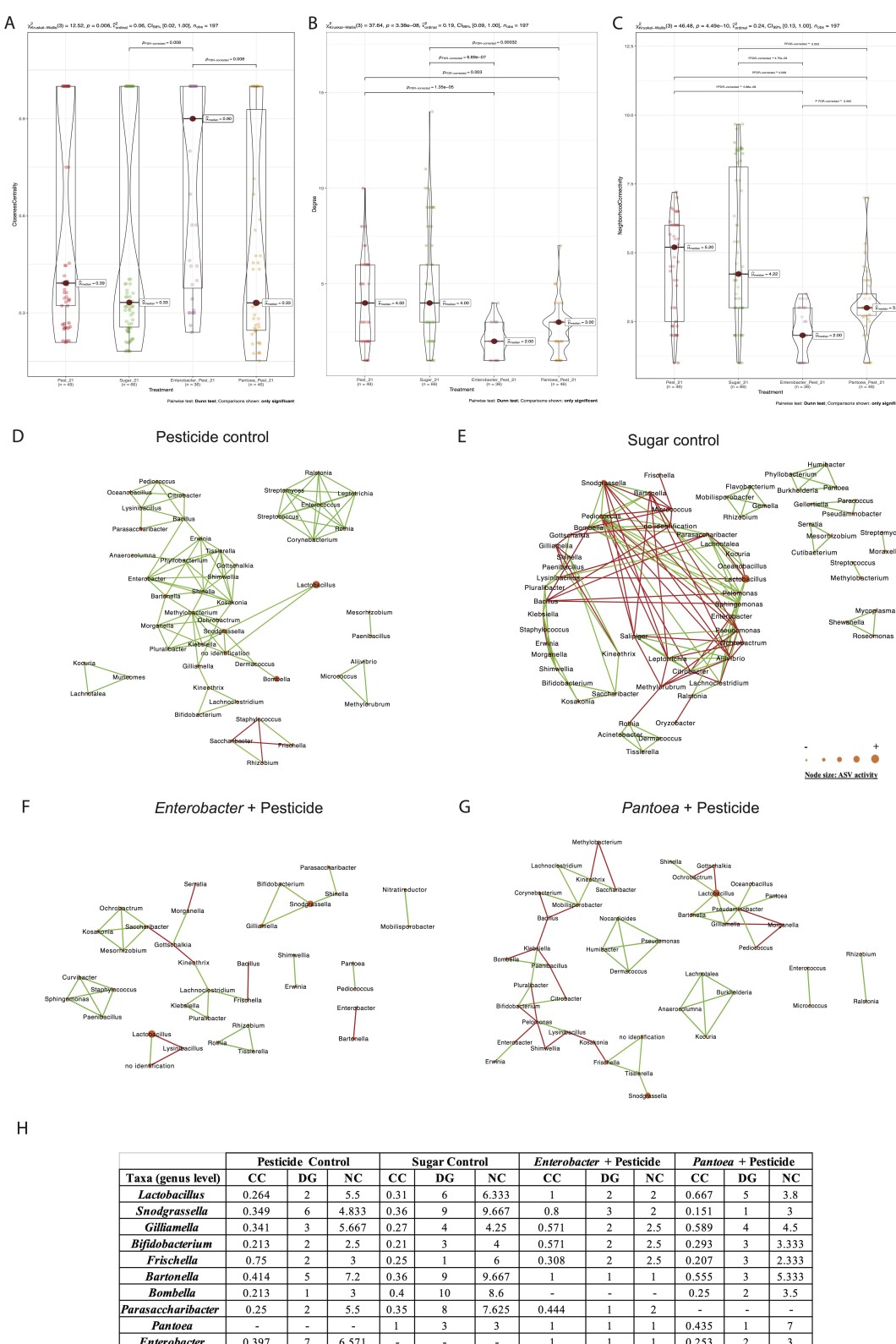

**FIG 3** Violin plot of (A) closeness centrality, (B) degree, and (C) neighborhood connectivity from taxa in ileum microbial networks; microbial networks for the ileum exposed at (D) 0.1 ppb (pesticide control), (E) 0 ppb (sugar control), (F) 0.1 ppb + *Enterobacter* (*Enterobacter* curative group), and (G) 0.1 ppb + *Pantoea*

**Fig 3 (Continued)**

(*Pantoea* curative group) that were generated based on pairwise correlations between the relative abundance of different bacterial genera. We used six replicates (five workers per replicate) per experimental condition. Each node represents a bacterial genus. The size of each node is proportional to the bacterial functional activity of each genus. The darker the node, the more interconnected it is. Each edge represents significant positive or negative Spearman correlation coefficients ($-1 \leq r \leq -0.4$) (negative, red) and ($0.4 \leq r \leq 1$) (positive, green); (FDR-adjusted $P$ value < 0.05). (H) Analysis of topological parameter values (CC, DG, NC) of core, non-core members, *Enterobacter*, and *Pantoea* ASVs under different experimental conditions (pesticide control, sugar control, *Enterobacter* + pesticide, *Pantoea* + pesticide). Pairwise Dunn test used. Only significant comparisons are shown.

with a significant increase for one *Bartonella apis* ASV, one *G. apicola* ASV, one *Lactobacillus mellis* ASV, two *Lactobacillus kimbladii* ASVs, three *Lactobacillus melliventris* ASVs, four *F. perrara* ASVs, and various unassigned ASVs; and with a decrease of activity for two *L. apis* ASVs, four *G. apicola* ASVs, and *S. alvi* ASVs. Regarding the non-core members, we observed a significant decrease for *P. apium*, *Bombella apis*, and two *F. perrara* ASVs; and a significant increase for one *Bartonella apis* ASV and four *F. perrara* ASV activity (Fig. 5). We observed a loss of connection with the *Lactobacillus* sp. but a gain of correlations with the *Bifidobacterium* sp. symbiont. In both conditions, *Pantoea* symbiont was characterized by a low functional activity and was among the highest CC values for both experimental conditions (Fig. 2D, G, and H).

In the ileum, comparisons between *Pantoea* curative group and pesticide control group showed that interacting ASV number (genus level) decreased from 49 to 46 (Fig. 3D and G; Table S2). Exposed ileum ASVs (0.1 ppb + *Pantoea* sp.) were significantly less connected (DG) relative to the pesticide control ileum network (0.1 ppb) ($P$ = 0.001; Fig. 3B, D, and G; Table S3). Significant positive correlations decreased from 97 to 40, and significant negative correlations increased from 3 to 20 (Table S3). Core members (such as *Gilliamella* sp., *Bifidobacterium* sp., and *Lactobacillus* sp.), non-core members (*Frischella* sp. and *Bombella* sp.), and low activity taxa already in interaction in the pesticide control ileum network gained negative correlations within the network exposed to *Pantoea*. Regarding neighborhood connectivity (NC), we observed significantly lower values for NC (0.1 ppb + *Pantoea* sp.) relative to the pesticide control ileum network ($P$ = 0.006; Fig. 3C and H). No significant difference was observed for the CC network parameter (Fig. 3A). In terms of ASV activity, we observed nearly the same changes observed in the midgut (Fig. 5). All core and non-core members were still the most active ASVs in correlations within the network (Fig. 3D, G, and H). We observed the gain of *Pantoea* genus in interaction inside the network, which was characterized by the highest NC value and a low functional activity within the network (Fig. 3D, G, and H).

In the rectum, comparisons between *Pantoea* curative group and pesticide control group showed that interacting ASV number (genus level) was stable, slightly decreased from 32 to 31 (Fig. 4D and G; Table S2). Exposed rectum ASVs (0.1 ppb + *Pantoea*) were significantly less connected (DG) relative to the pesticide control rectum network (0.1 ppb) ($P$ = 0.001; Fig. 4B, D, and G; Table S3). Significant positive correlations decreased from 137 to 33, and 13 significant negative correlations appeared (Fig. 4D and G; Table S3). Non-core members, such as the *Frischella* sp., and low activity taxa already in interaction in the pesticide control rectum network gained negative correlations within the network exposed to *Pantoea*. Regarding neighborhood connectivity (NC), we observed significantly lower values for NC (0.1 ppb + *Pantoea*) relative to the pesticide control rectum network ($P$ = 3.64 e-05; Fig. 4C and H). Regarding closeness centrality (CC), we observed lower values for CC (0.1 ppb + *Pantoea*) relative to the pesticide control rectum network ($P$ = 0.000154; Fig. 4A and H). However, the higher CC values for (0.1 ppb + *Pantoea*) were corresponding to the *Bifidobacterium* sp., the *Bartonella* sp., and low activity taxa, whereas for the pesticide control rectum network, the highest CC values were corresponding only to low activity taxa (Table S4 and S5). Rectum was less impacted by exposure to clothianidin (0.1 ppb) and *Pantoea*. All core and non-core members were still the most active ASVs in correlations within the network (Fig. 4H). We observed a significant increase of activity for the following core: *L. apis*, *L. melliventris*, and *G. apicola* symbiont. *Snodgrassella* sp. ASV activity showed to be impacted (increase

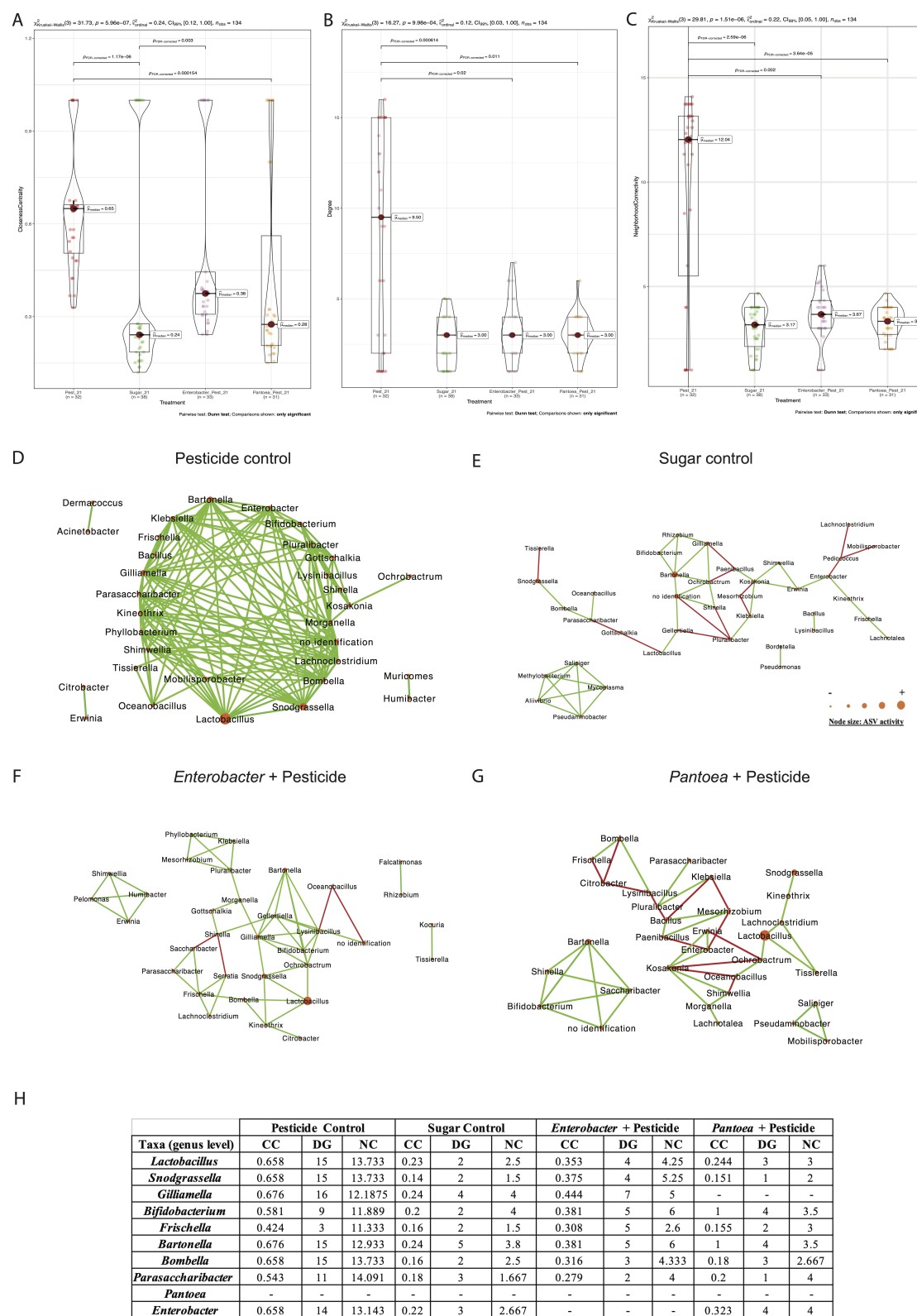

**FIG 4** Violin plot of (A) closeness centrality, (B) degree, and (C) neighborhood connectivity from taxa in rectum microbial networks; microbial networks for the rectum exposed at (D) 0.1 ppb (pesticide control), (E) 0 ppb (sugar control), (F) 0.1 ppb + *Enterobacter* (*Enterobacter* curative group), and (G) 0.1 ppb + *Pantoea* (*Pantoea* curative group) that were generated based on pairwise correlations between the relative abundance of different bacterial genera. We used six replicates (Continued on next page)

Fig 4 (Continued)

(five workers per replicate) per experimental condition. Each node represents a bacterial genus. The size of each node is proportional to the bacterial functional activity of each genus. The darker the node, the more interconnected it is. Each edge represents significant positive or negative Spearman correlation coefficients ($-1 \leq r \leq -0.4$) (negative, red) and ($0.4 \leq r \leq 1$) (positive, green); (FDR-adjusted $P$ value < 0.05). (H) Analysis of topological parameter values (CC, DG, NC) of core, non-core members, *Enterobacter*, and *Pantoea* ASVs under different experimental conditions (pesticide control, sugar control, *Enterobacter* + pesticide, *Pantoea* + pesticide). Pairwise Dunn test used. Only significant comparisons are shown.

and/or decrease) only with *Pantoea* sp. exposure in the three gut sections. Regarding the non-core member and unassigned ASVs, we observed a decrease of their ASV activity (Fig. 5).

To encapsulate the main outcomes, the 21-day exposure to putative beneficial microbes, including *Enterobacter* and *Pantoea* strains, led to distinct reshaping of honeybee gut microbiota dynamics across various gut sections. Exposure to *Enterobacter* resulted in reduced interactions among ASVs in the midgut and ileum, with increased connectivity and positive correlations observed specifically in the midgut. Conversely, the ileum exhibited decreased interactions and connectivity, accompanied by changes in correlation patterns and decreased neighborhood connectivity. In the rectum, ASV interactions remained stable but with decreased connectivity and positive correlations, notably impacting non-core members. On the other hand, exposure to *Pantoea* led to decreased interactions and connectivity in the midgut and ileum, accompanied by reduced closeness centrality. Similar trends were observed in the rectum, with decreased ASV interactions, connectivity, and positive correlations, particularly affecting core members like *Bifidobacterium* and *Bartonella*. These findings underscore the differential effects of *Enterobacter* and *Pantoea* strains on gut microbiota composition and

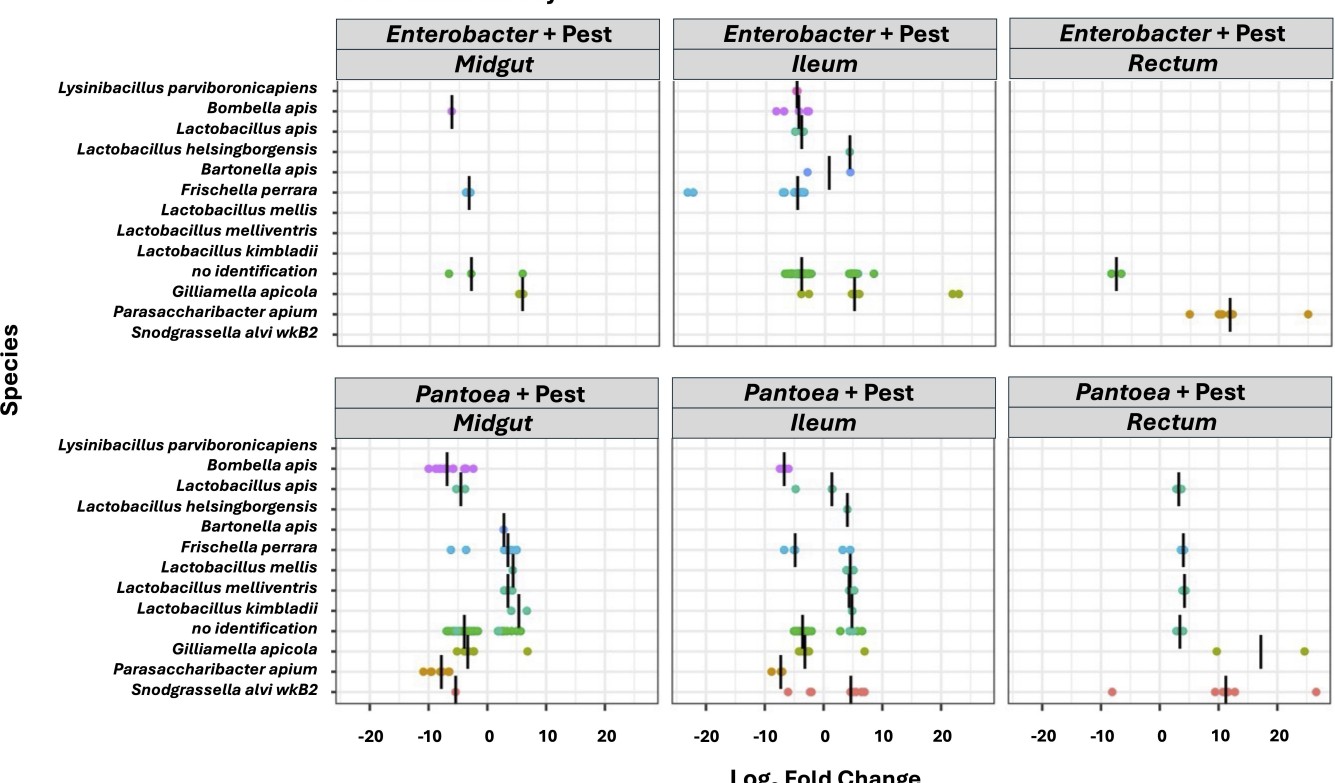

**FIG 5** Differential activity of ASVs (total) significantly different ($P$ < 0.05) between each experimental gut section (ileum, midgut, and rectum) exposed to pesticide (0.1 ppb) and *Enterobacter* (top figures) or *Pantoea* (bottom figures) (curative effect) relative to the control pesticide at T21 (six replicates per experimental condition). Negative fold change scores (log2) indicate genera with decreased activity in curative effect samples, and positive fold change scores indicate genera with increased activity. Each point represents an ASV, and only the significant difference in genera activity (adjusted $P$ < 0.05) is shown.

network dynamics, suggesting potential implications for gut microbiota homeostasis in honeybees throughout their digestive tract.

## DISCUSSION

This present work represents an *in vivo* experiment studying the impact of two bacterial strains isolated from the gut of healthy honeybees belonging to *Enterobacter* and *Pantoea* genera as beneficial microbes to mitigate honeybee gut dysbiosis induced by clothianidin. Our results suggest that one of our bacterial strains (*Pantoea* sp.) may be promising in mitigating clothianidin adverse effects on honeybees. At the outset, the administration of the *Pantoea* sp. strain as a curative measure significantly improved the survival of honeybees exposed to clothianidin, surpassing the survival of bees in the sugar control group (i.e., those supplemented with sugar in a 1:1 ratio). On the other hand, although curative administration of the *Enterobacter* sp. appeared to alleviate the clothianidin-induced dysbiosis observed in the midgut, it did not improve bee survival when compared with the pesticide group. Our main goal was to assess the extent to which the administration of bacterial strains would (i) reestablish the bee gut eubiosis disrupted by clothianidin exposure (33) and (ii) induce a beneficial impact on host health (58). We tested whether administration of both of these bacterial strains would have a positive impact on the structure of the honeybee gut microbiota by (i) increasing and/or maintaining mutualistic interactions with beneficial strains (59); (ii) decreasing and/or limiting antagonistic interactions with opportunistic pathogens (60); and *(*iii) bringing the gut microbiota structure (curative group) either to (i) the original eubiotic state (as the sugar control group)We used the rco (58) or to (ii) a new microbial equilibrium, as observed in yellow perch exposed to sublethal doses of cadmium chloride (57).

### Long-term clothianidin exposure differentially altered the honeybee gut bacterial communities

Studies show that exposure to xenobiotics and antibiotics can affect microbiota composition differently in terms of taxonomic richness and structure. Some researchers (61–63) observed a reduction in gut microbial species diversity; whereas others (33) found an increase in gut microbiota alpha diversity in bees exposed to clothianidin (0.1 ppb). However, Raymann et al. (64) did not find any disturbance in the gut microbiota after imidacloprid exposure. These varying observations were based on different experimental factors, including the type of chemical, exposure time (65), and chemical concentrations (33, 66). However, our study used the same experimental conditions (long-term exposure) and pesticide concentration (0.1 ppb), as in El Khoury et al. (33), except the exposure in the present study began at the onset of the experiment (Day 0), whereas in El Khoury et al. (33), clothianidin exposure began on Day 3. We looked at how pesticide exposure affects the gut microbiota by comparing network patterns of experimental groups at two different times (T7 and T21) under the same conditions as El Khoury et al. (33).

Compared to the control group (0 ppb), the 0.1 ppb group showed increased negative interactions within the midgut section at T7 (33) and T21. However, in the ileum section, negative interactions decreased slightly at T7 and dropped significantly at T21 for the 0.1 ppb group. At T7, the 0.1 ppb group had very few interactions within the rectum section compared to the 0 ppb group, but at T21, the 0.1 ppb group had many interactions and exhibited no negative interactions compared to the 0 ppb group. Overall, the midgut and ileum exhibited a similar pattern at T7 and T21 when compared to their respective control groups. The midgut, essential for xenobiotic detoxification (67, 68), exhibits a diminishing dysbiosis pattern from T7 to T21 when chronically exposed to 0.1 ppb, suggesting potential midgut microbiota resilience. This interpretation is based on the hypothesis that the rise of negative interactions is a hallmark of dysbiosis (33, 57, 69). Our research suggests that clothianidin exposure alone did not solely cause dysbiosis in the two other gut sections. It is probable that factors such as restricted

feeding of sugar syrup and cage confinement contributed to stress in all three gut sections.

We observed that caged bees host a greater bacterial diversity than in the hive, likely because bees will not defecate in cages (70), potentially creating high dysbiosis in both ileum and rectum, as observed in 0 ppb control groups at T21. The ileum and rectum handle non-digested pollen components and have the highest bacterial density, whereas the midgut breaks down simple sugars (71). Caged bees, which only had access to sugar syrup, may have induced bacteria in the ileum and rectum to be more competitive due to lack of food sources. This could have led to more antagonistic interactions in the sugar control groups. By providing nutritional metabolites to ileum and rectum bacterial communities, 0.1 ppb of clothianidin degraded by endogenous bacteria (26) potentially mitigated dysbiosis in the ileum and rectum.

## The two honeybee gut bacterial strains induced different gut microbiota signatures

Our work shows that exposure to the *Enterobacter* sp. and *Pantoea* sp. exhibited different signatures in gut microbiota structure depending on the administered bacterial strain and gut section. In all gut sections of both curative groups, bacterial networks exhibited principally (i) changes in interactions, (ii) a decrease of interacting ASVs (except for the *Enterobacter* sp. in the rectum), (iii) a decrease of core and non-core members' activity, (iv) a rise of beneficial ASV members, and (v) the decrease in the values of network topological parameters (CC, DG, NC), relative to the pesticide control group. Number and sign (+/−) of active ASV interactions are used to assess the impact of environmental changes on gut microbiota homeostasis (72–74), where a rise of negative interactions is interpreted as a dysbiosis signature (33, 57, 69).

In curative groups, a general decrease in positive correlations was compensated by an increase of negative correlations (in both ileum and rectum), whereas it remained stable in the midgut for the *Pantoea* sp. and exhibited a rise of positive correlations for the *Enterobacter* sp. with no negative correlation (Table S3). Conversely, a rise of negative correlations indicates how microbes within networks may compete or prey on each other (73). As discussed in the previous section, ileum and rectum bacteria are specialized in degrading complex molecules such as pollen proteins for instance. Bacteria in curative groups likely benefited from clothianidin degradation, unlike the sugar control group that exhibited more negative interactions among bacteria. This suggests that the breakdown of clothianidin into various metabolites may have decreased resource competition in the ileum and rectum sections.

Contrastingly, in the midgut curative group, the *Enterobacter* sp. showed increased positive correlations (x3.5) with bee core members like the *Lactobacillus* sp. and *Gilliamella* sp., as well as between the *Bifidobacterium* sp. and *Gilliamella* sp. The *Bifidobacterium* sp. and *Gilliamella* sp. are known to enhance honeybee immunity (71, 75). *Gilliamella* sp. activity decreased with clothianidin (0.1 ppb) in the present study and in El Khoury et al. (33), whereas bacterial strains administration in the curative group restored *Gilliamella* sp. activity. Furthermore, no negative correlation between the interacting microbes were detected. Then, as exposed in Bonilla-Rosso and Engel (60), we observed the loss of interactions with microbes known as pathogens such as the *Staphylococcus* sp., and a gain of connections with low activity taxa (for the *Enterobacter* sp., curative group) relative to the midgut pesticide control group. These results might suggest that administering *Enterobacter* had a positive effect on the midgut curative network, but did not lead to improved survival. For *Pantoea* administration, which led to better survival when exposed to clothianidin, negative correlations remained similar to the pesticide control group, but the number of network modules dropped from eight to three, indicating a much more connected network for the *Pantoea* curative group.

A decreasing number of interacting ASVs was observed in the ileum and rectum for both curative groups relative to the pesticide control group (Table S2). Once again, ileum and rectum bacteria in the pesticide control group may have benefited from

clothianidin and its metabolites following degradation by several bacterial strains (33), therefore reducing competition and promoting syntrophic relationships between strains degrading clothianidin and others using derived metabolites as nutritional resources. It is possible that administration of two bacterial strains able to degrade clothianidin *in vitro* disturbed the endogenous capacity of the gut microbiota to degrade the pesticide molecule and/or the nutritional properties of secondary metabolites resulting from its degradation. Nevertheless, the different performances recorded in terms of honeybee survival between *Pantoea* curative group relative to both *Enterobacter* curative and pesticide control groups may be due to varying toxicity levels of clothianidin metabolites produced in each group. To that respect, clothianidin might have been transformed into more toxic substances such in both *Enterobacter* curative and pesticide control groups, compared to the *Pantoea* curative group.

## Honeybee gut bacterial strains induced a beneficial gut microbiota signature

A general tendency of decreasing network parameters (CC, NC, and DG) was observed in both curative groups relative to what occurred in the pesticide control group. When focusing on the midgut, which is known to be involved in xenobiotic detoxification (67, 68), the lack of significant differences in terms of CC between *Pantoea* curative group and sugar control group on one hand, and between *Enterobacter* curative group and pesticide control group on the other (Fig. S1) may suggest convergences between *Pantoea* and sugar control group and between *Enterobacter* and pesticide control group respectively. Closeness centrality (CC) measures the average influence of each node in the overall network. Bacteria with beneficial properties (i.e., inducing beneficial effect on the host) such as the *Bifidobacterium* sp., the *Bartonella* sp., and some unknown bacterial genera with low activity were characterized with high CC value supporting their strong involvement in the curative *Pantoea* group interacting network. Interestingly, it is their administration that has rewired the interacting network in a curative context. To this end, both activity levels and network parameters were compared between the four experimental groups: sugar control, pesticide control, and both *Enterobacter* and *Pantoea* curative groups because without these bacterial treatment, other endogenous strains belonging either to *Enterobacter* or *Pantoea* genera were active in these four experimental groups.

Relative to the sugar control group, endogenous *Enterobacter* strain activity was slightly higher in the midgut and lower in both ileum and rectum (Fig. 2H). Regarding interactions with other strains in the pesticide control, all network parameters strongly increased in the midgut and to some extent in the rectum, whereas they remained stable in the ileum (Fig. 2H, 3H, and 4H). When *Pantoea* was administered, it interacted more with other pesticide-controlling bacteria in the midgut but not in the ileum. However, it was not found in the rectum of either sugar or pesticide control groups. Both administered honeybee gut microbial strains seem to have responded to clothianidin exposure by interacting more with the bacterial community, especially in the midgut, which is known to help detoxify harmful substances (67, 68).

In the midgut of the *Enterobacter*-treated honeybees, the combined *Enterobacter* community was less active, relative to both sugar control and pesticide control, therefore translating into a neutralizing or competing effect on the endogenous *Enterobacter* community. Then, both CC and DG network parameters remain in the range of their respective values in the sugar control and pesticide control, whereas NC values are higher than in both controls (Fig. 2H), indicating a positive effect of administering a related strain to the endogenous *Enterobacter* community in terms of interactions with the other bacterial members. Then, all network parameters were lower (ranging between a half and a third) than the values recorded in both sugar control and pesticide control (Fig. 3H), therefore translating into a neutralizing or competing effect on the endogenous *Enterobacter* community. Finally, *Enterobacter* was not detected in the rectum, in contrast to both control groups (Fig. 4H), which indicates a neutralizing or competing effect of administering *Enterobacter* on the endogenous *Enterobacter* community. Overall,

administrating *Enterobacter* appears to slightly favor the capacity of the endogenous *Enterobacter* community to interact with other active bacterial members in the midgut, but partially neutralizes the endogenous *Enterobacter* community in both ileum and rectum.

In the *Pantoea*-treated group in the midgut, the resulting combination of both endogenous and administrated *Pantoea* strains translated into higher activity (x9), lower CC, relative to both sugar and pesticide control groups (Fig. 2H). Then, NC and DG were similar to the pesticide control, whereas higher (x4) relative to the sugar control. In the ileum, the combined *Pantoea* strains translated into higher activity (x13) and higher NC (x2.3), relative to both sucrose and pesticide control groups (Fig. 3H). Then, CC and DG were below values recorded in the sugar control, but quite higher than in the pesticide control, the latest displaying no *Pantoea* interaction. Finally, no *Pantoea* interaction was detected in the rectum (Fig. 4). Overall, *Pantoea* administration strongly supports the activity and interaction of the endogenous *Pantoea* community (i.e., high NC and DG values) with other bacteria in the midgut, as well as in the ileum.

Altogether, *Enterobacter* and *Pantoea* bacterial strains administration exerted differential influence in the capacity of their endogenous correlatives to interact with the other bacterial strains in all gut sections. These differences in interaction patterns should be paralleled by the ability of the bacterial strain *Pantoea* to improve survival in a curative setting. In this regard, *Pantoea* curative administration showed the highest survival rates among all groups (Fig. 1), possibly because it improved both activity and connectivity of its endogenous community. This suggests that enhancing the *Pantoea* community's activity and connectivity contributes to its beneficial effects.

In addition to inducing a positive impact on both activity and connectivity of endogenous *Pantoea* community, administered *Pantoea* improved honeybee gut microbiota by increasing beneficial bacterial activity, including core and non-core members involving bacteria known for their beneficial properties (Fig. 5). For instance, in all the three gut sections, *Lactobacillus milliventris*, *F. perrara*, *G. apicola*, and *S. alvi* were significantly more active relative to the pesticide control group. Majority of previous work on honeybees focused their analyses on core, non-core bee gut members, and bacterial genera such as the *Lactobacillus* sp. (76), *G. apicola,* and *S. alvi* (18, 77) that are well known for their beneficial properties on the host. For instance, honeybees fed with *Lactobacillus* spp. exhibited a diminished level of *Vairimorpha* (*Nosema*) *ceranae* infection (78), whereas *P. apium* and the *Bacillus* sp. improved bee survivability in *V. ceranae* infection experiments (48). Administering endogenous bee microbes in honeybees exposed to tylosin showed enhanced survival in both prophylactic and infected groups with *Serratia marcescens* (79).

Commercial probiotics (i.e., microbes inoculation inducing beneficial effect on the host), specifically the *Lactobacillus* sp., fed to *V. ceranae*-infected bees resulted in reduced survival (45), whereas El Khoury et al. (48) observed an enhancement of bee survival following Bactocell and Levucell SB commercial probiotics administration. Moreover, in a neonicotinoid context, administration of probiotic *Lactobacillus plantarum* to imidacloprid-exposed *Drosophila melanogaster* helped them to counter *Serratia* sp. infection (39). These findings support the potential for endogenous bee gut microbes in preventing (i) infection by pathogens (79) and (ii) neonicotinoid-induced infection susceptibility (39). Beneficial microbes feeding in honeybees has been effective in boosting immune key genes and promoting honeybee survival in countering pathogens (40, 79, 80). Whereas, beneficial properties of both *Enterobacter* and *Pantoea* genera were not characterized so far in honeybees, but strains belonging to the same genera showed beneficial effects in other insect species (49–52).

*Enterobacter* sp. and *Pantoea* sp. supplementation in honeybees exposed to clothianidin resulted in an increase in bacterial genera with beneficial properties such as *Bifidobacterium*, *Lactobacillus*, and the *Parasaccharibacter* sp., which had positive interactions in curative bacterial networks. Beneficial microorganisms have shown to help in preventing gut dysbiosis by altering gut microbiota diversity (58). *Lactobacillus*

and the *Bifidobacterium* sp. (LABs) are known to be involved in bee health (81, 82) and in colonization resistance against pathogens. Moreover, *Lactobacillus* sp. administration has shown to (i) be involved in a reduction of chalkbrood disease (83); (ii) act as an antagonist against *Paenibacillus larvae* (84); (iii) reduce bee mortality induced by *P. larvae*; and (iv) decrease *V. ceranae* sporulation (85). An increase of *Acetobacteraceae* and *Bifidobacterium* species abundance in the bee gut microbiota was also observed in bees fed with *Bifidobacterium* and *Lactobacillus* sp. probiotic candidates, and are known to be involved in bee nutrition and protection (40). Altogether, these studies not only endorse the beneficial impact of endogenous bee microbes on unhealthy honeybees but also suggest how beneficial-based prophylaxis depends on the bacterial strains used. Interestingly, either with administration of beneficial microbes or not (33), 0.1 ppb clothianidin exposure in both experimental conditions resulted in an increase of ASVs with beneficial properties, which could alleviate the pesticide's harmful effects on honeybees.

## Environmental honeybee gut microbes in honeybee nutrition

Our work sheds light on the positive impact of environmental honeybee gut microbes on each gut microbiome section, including interactions between core and non-core microorganisms. Administering endogenous microbes to honeybees can positively affect their gut microbiota in response to pesticide exposure, promoting an alternate eubiotic equilibrium. Ileum and rectum networks had unexpected dysbiosis in the sugar control and both curative groups, unlike in the pesticide control. Ileum and rectum bacteria may have benefited from clothianidin molecules and their metabolites as nutritional resources, in contrast to the sugar control group. Nevertheless, the toxicity of clothianidin metabolites would have been lower in the *Pantoea* sp. curative group, when considering survival data. Overall, our work supports the use of endogenous gut bacterial strains in honeybee nutrition *in vivo* and should be validated for their ability to mitigate the negative impacts of neonicotinoids on honeybee colonies *in situ*.

## MATERIAL AND METHODS

### Bacterial strain candidates

Two bacterial strains belonging to *Enterobacter* and *Pantoea* genera were isolated from the whole honeybee gut, demonstrated *in vitro* beneficial properties regarding their ability to degrade clothianidin, and were therefore selected for this work (26).

### Chemical compound and quantification method used

Clothianidin (CAS Number 210880-92-5) was supplied by Sigma-Aldrich Inc. (Oakville, Ontario, Canada). Clothianidin titers were measured before use and were obtained by liquid chromatography/tandem mass spectrometry (LC-MS/MS) at the INRS (Institut National de la Recherche Scientifique, Québec, Canada) as in El Khoury et al. (26, 33). Briefly, to quantify clothianidin concentration, we employed a modified QuEChERS method that combines chromatography and tandem mass spectrometry. Methanol (MeOH) was used as a stock standard for calibration and recovery determination, at 4°C in a dark room. We also used atrazine-D5 as an internal standard, both of which were purchased from CDN Isotopes (Pointe, Québec, Canada). A mixture of salt (magnesium sulfate [$MgSO_4$, 4 g], sodium chloride [NaCl, 1 g], sodium citrate dihydrate [1 g], and disodium citrate sesquihydrate [0.5 g]) was added to each sample (five bees + distilled water [1 mL] + acetonitrile [1.5 mL]) to quantify, agitated for 15 minutes, and centrifuged at $3,000 \times g$ for 5 minutes at room temperature. Then, we transferred the supernatant (500 µL) to a new culture tube, evaporated the mixture using a nitrogen evaporator at 40°C, and then rehydrated it with a solution of water and methanol, adding atrazine-D5 as an internal standard. We transferred 100 µL of the resulting solution to a new tube for analysis with LC-MS/MS to quantify clothianidin pesticide. We used a TSQ Quantum Access MAX Triple Quadrupole Mass Spectrometer (Thermo Fisher Scientific, San Jose,

CA, USA) and a Hypersil Gold aQ column (Thermo Fisher Scientific) to analyze the sample with liquid chromatography. A linear gradient from 1.1 to 3 minutes was used to elute clothianidin, and then returned to the initial conditions for an additional few more minutes. An electrospray ionization source in a positive mode and a triple quadrupole mass spectrometer were used to detect the ions. The ion tube was heated at 350°C. Clothianidin and atrazine-D5 were characterized by their retention times and quantification transitions, and we used Xcalibur Software (Thermo Fisher Scientific) to quantify them. We determined the concentration of clothianidin in the sample by comparing the area ratio of the peaks for clothianidin and the initial standard to a calibration curve for the clothianidin standard.

## Experimental setup

We based our experiment on a previous protocol (33). Our *in vivo* experiment was performed between July and August 2018, on newly emerged honeybees at the Centre de Recherche en Sciences Animales de Deschambault (CRSAD, Québec, Canada). All honeybees used in this study originated from five European honeybee colonies (*Apis mellifera* L.) headed by queen sisters. Newly emerging honeybees were obtained as described by Williams et al. (86), using a "nursery colony" made of a Langstroth hive body with five combs of capped brood (purple eye), one frame of honey and pollen, and some adherent nurse bees (approximately 20–30 nurse bees per frame) from the original colonies. The nursery colony was incubated at 32°C and 55% relative humidity in a Model 3040 apparatus (Forma Scientific Inc., Marietta, OH, USA) for 6 days. Young honeybees emerged in nursery colonies and were kept there for 4–6 days to ensure optimal microbiota acquisition/colonization from nurse bees (54). After this incubation period, young honeybees were hand collected and placed in plexiglass cages. Two hundred honeybees were randomly distributed in each cage (three cages per group) for a total of 3,600 bees in all groups for a total of six groups. Each cage consisted of a plexiglass structure (10 × 10 × 10 cm) adapted from Evans et al. (87) with an inverted sterile syringe (20 mL, BD, Franklin Lakes, NJ, USA) containing 50%$_{wt/vol}$ (1:1) sugar syrup (sugar diluted in distilled water). Cages were kept in an environmentally controlled room (30°C ± 1°C and 50% ± 5% relative humidity) in darkness for the duration of the experiment (28 days). Cages were randomly distributed between the six groups, and syrup solutions were prepared and administered as shown in Table 1 (four control groups: sugar control, pesticide control, *Enterobacter* control, *Pantoea* control; and two curative groups: *Enterobacter* + pesticide and *Pantoea* + pesticide). Honeybees' beneficial microbe candidates (one of the two strains of *Enterobacter* and *Pantoea* genera) were fed a dose of $10^4$ CFU/mL mixed in sugar syrup (1:1); honeybees exposed to pesticide were exposed to a sublethal clothianidin concentration of 0.1 ppb. For each treatment (three cages per group), we started to feed 200 honeybees per cage from Day 1 (T1). All treatments began at T1 until T28. Each cage received a daily supply of fresh sugar solution. Every day, mortality was recorded in each cage. At T21, five honeybees were randomly sampled from each cage and were stored (on the spot) at −80°C for metataxonomic analysis of the honeybee gut microbiota.

TABLE 1  Description of experimental groups[a]

| Group | Sample size (cage) | Pesticide | CFUs | Pesticide exposure | CFU administration | 50%$_{wt/vol}$ sugar syrup administration |
|---|---|---|---|---|---|---|
| Sugar control | 3 | | | | | T1-T28 |
| Pesticide control | 3 | 0.1 ppb | | T1-T28 | T1-T28 | T1-T28 |
| *Enterobacter* control | 3 | | 10^4 | T1-T28 | T1-T28 | T1-T28 |
| *Enterobacter* + pesticide | 3 | 0.1 ppb | 10^4 | T1-T28 | T1-T28 | T1-T28 |
| *Pantoea* control | 3 | | 10^4 | T1-T28 | T1-T28 | T1-T28 |
| *Pantoea* + pesticide | 3 | 0.1 ppb | 10^4 | T1-T28 | T1-T28 | T1-T28 |

[a]Each group consists of three cages, 200 honeybees per cage for a total of 600 honeybees.

## Survival analysis

To estimate honeybee survival rates, we used the Kaplan-Meier estimator (88) implemented in the survival R package (version 3.2.7) (89). Statistically significant risk differences between treatments: (i) unexposed honeybees (sugar control) versus (1) honeybees exposed to clothianidin (clothianidin effect), (2) bacterial strains belonging to *Enterobacter* and *Pantoea* genera (prophylactic effect [microbial control groups]); (ii) honeybees exposed to clothianidin (control pesticide) versus honeybees exposed to the combination (strain of *Enterobacter* or *Pantoea* genus + clothianidin) (curative effect); and (iii) unexposed honeybees (sugar control) versus curative effect (bacterial strain of *Enterobacter* or *Pantoea* genus + clothianidin). All comparisons were analyzed with a Cox's proportional hazards regression using the CoxME model (with "cages" as random effect) implemented in the survival R package (89). Significant differences were calculated using a multiple comparison post-CoxME, and $P$ values were adjusted with the Tukey test.

## Honeybee dissection

Each gut sample was split in three sections (midgut, ileum, and rectum). First, honeybee samples stored at −80°C were left standing on ice. We performed cuticle sterilization to avoid microbial contamination during gut dissection. Briefly, the honeybees were washed using a diluted bleach solution (1:100) for 2 minutes. Then, we rinsed each honeybee separately three times in clean distilled water to remove bleach residues, followed by centrifugation for 45 seconds at 10,000 × $g$ at 20°C to remove all remaining residues at the bottom of the tubes. Then, we used two replicates of five bees per cage (10 honeybees in total per cage) for subsequent RNA extraction.

## RNA extraction, 16S rRNA gene amplicon library construction and sequencing

RNA extraction, cDNA synthesis, two-step 16S rRNA gene amplicon library preparation, and paired-end Illumina sequencing were performed as described by El Khoury et al. (33). Briefly, at T21, tissue samples were taken from three honeybee gut regions (midgut, ileum, and rectum). Tissue samples from the same cage (five gut sections isolated from five honeybees per replicate and per cage) were pooled for RNA extraction, and RNA was extracted using the TriReagent method (Ambion, Thermo Fisher Scientific). Then, the qScriptTMcDNA SuperMix method (QuantaBio, VWR, Beverly, MA, USA) was used to reverse transcribe RNA samples into complementary DNA (cDNA) according to the manufacturer's instructions (90). Following that, a two-step dual indexing technique was used to produce partial 16S rRNA amplicons of the hypervariable V3-V4 regions (33). The barcoded amplicons were pooled in equimolar concentrations and were sequenced using Illumina MiSeq paired-end technology (2 × 300 bases) at Laval University's "Plate-forme d'Analyses Génomiques." As a calibration control, we used 15%–20% of the PhiX control v3 Library (MiSeq Reagent kit v3 600 cycles PE, Illumina, Inc., San Diego, CA, USA).

## Bioinformatics

### Sequence clustering

In total, 108 samples (2 replicates of 5 honeybees × 3 gut sections × 3 cages per group × 6 treatments × 1 time point = T21) were sequenced individually. Raw sequences from all samples were checked for base calling quality using FastQC (https://www.bioinformatics.babraham.ac.uk/projects/fastqc/). For the following steps, we used exactly the same methodology as in El Khoury et al. (33) in order to compare both studies. The reads were processed using the dada2 pipeline (version 1.12) following the method of Callahan et al. (2016) (91). Before analyzing the data, the quality of the reads was checked using the filterAndTrim function with specific settings: a truncation length of

270, a Phred score threshold of 2 for total read removal, and a maximum expected error of 2 for forward reads and 4 for reverse reads. Prior to analysis, we obtained a total of 19,430,348 sequences from which we kept a total of 14,975,759 reads after filtration. After filtration, reads were then used to learn the error rate, remove duplicates, and identify the ASVs, which was accomplished by using the learnErrors, derepFastq, and dada functions, respectively.

## Taxonomic identity

Taxa were classified by using blast matches from the NCBI 16S Microbial database (92). Matches with an identity score above 98% were assigned a taxonomic identity. For sequences without matches above this threshold, we used a last common ancestor (LCA) method based on the top 50 matches to assign taxonomy. We were inspired by the LCA algorithm used in MEGAN (93).

## Differential activity analysis

Differential activity analysis, i.e., differential expression of the 16S rRNA gene, was performed with the DESeq2 package (v.1.30.0) (94) to determine statistically significant differences for ASVs in terms of activity ($P < 0.05$) between pesticide control versus PC1 (or PC2) combined with the pesticide to analyze the curative effect of both probiotics tested on each gut section at time T21.

## Microbial network analysis

Co-expression networks were constructed by using Rstudio (version 1.13.1093). Significant correlations were highlighted between taxa in each honeybee gut section under different experimental conditions. Hmisc R package (version 4.2-0) (95) was used to construct correlation matrices with $P$ values corrected using the false discovery rate (FDR) method (96). We used the rcorr() function in the Hmisc package to calculate Spearman's rank correlation coefficients and their associated $P$ values for all possible pairs of ASVs. Pairs of ASVs with correlation coefficients greater than or equal to 0.4 or less than or equal to −0.4, and $P$ values less than 0.05 were considered significant (33). The methodology using Spearman coefficients to infer correlation is trustworthy and similar to modern mutual information methods (57). Twelve microbial networks were created by calculating pairwise correlations between the functional activity of each taxon at the genus level. Microbial networks were visualized using Cytoscape software version 3.8.2 (97). In each network, nodes represent bacterial genera, with the node size proportional to the genus functional activity, and the node color representing how interconnected it is in the network. Bee gut taxa that occurred in most replicates ($n >$ 3 out of 6 per experimental condition) were included in network interpretation. Genera with low activity (<0.01% of the total sample activity) that occurred in few samples ($n < 3$ out of 6 per experimental condition) were considered as having low activity. For network interpretation, we analyzed three measures of network topology obtained using the Network Analyzer function through Cytoscape: closeness centrality (CC), degree (DG), and neighborhood centrality (NC). CC measures a node's centrality and ability to interact with other nodes. DG measures local communication activity (98, 99), and NC reflects a node's impact on overall network dynamics (100, 101).

## ACKNOWLEDGMENTS

The authors would like to express their gratitude to the Centre de recherche en sciences animales de Deschambault (CRSAD) and the Institut de Biologie Intégrative et des Systèmes (IBIS). They are grateful to Emile Houle for all the technical help in the field; Andrée Rousseau for taking care of the nursery colony until bees' reception; and Sidki Bouslama for advice in bioinformatics. A final thanks to Dr. Bachar Cheaib for all the discussions regarding this project.

This research was funded by Innov'action program of MAPAQ and Agri-Food Canada (#IA115285).

Conceptualization, S.E.K., P.G., and N.D.; experimental work, S.E.K.; amplicon library preparation, S.E.K. and P.L.M.; clothianidin quantification, M.S.; data analysis, S.E.K. and J.G.; writing—original draft, S.E.K.; writing—review and editing, S.E.K., P.G., and N.D.; funding acquisition, P.G. and N.D. All authors have read and agreed to the published version of the manuscript.

## AUTHOR AFFILIATIONS

[1]Université Laval, Institut de Biologie Intégrative et des Systèmes (IBIS), Québec, Canada
[2]Département de Biologie, Université Laval, Québec, Canada
[3]INRS, Institut National de la Recherche Scientifique, Québec, Canada

## PRESENT ADDRESS

Sarah El Khoury, Department of Integrative Biology, University of California, Berkeley, California, USA

## AUTHOR ORCIDs

Sarah El Khoury  http://orcid.org/0000-0003-1846-1043
Jeff Gauthier  http://orcid.org/0000-0002-2523-2698
Nicolas Derome  http://orcid.org/0000-0002-2509-6104

## FUNDING

| Funder | Grant(s) | Author(s) |
|---|---|---|
| Ministère de l'Agriculture, des Pêcheries et de l'Alimentation (MAPAQ) | IA115285 | Pierre Giovenazzo |
| | | Nicolas Derome |
| Canadian Government | Agriculture and Agri-Food Canada (AAFC) | IA115285 | Pierre Luc Mercier |
| | | Nicolas Derome |

## AUTHOR CONTRIBUTIONS

Sarah El Khoury, Conceptualization, Data curation, Formal analysis, Investigation, Methodology, Supervision, Validation, Visualization, Writing – original draft, Writing – review and editing | Jeff Gauthier, Formal analysis | Pierre Luc Mercier, Methodology | Stéphane Moïse, Methodology | Pierre Giovenazzo, Conceptualization, Funding acquisition, Investigation, Supervision, Validation, Writing – review and editing | Nicolas Derome, Conceptualization, Funding acquisition, Investigation, Supervision, Validation, Writing – review and editing

## DATA AVAILABILITY

The raw sequence reads analyzed in the current study are available in the NCBI BioProject ID repository under the number PRJNA678327.

## ADDITIONAL FILES

The following material is available online.

### Supplemental Material

**Supplemental material (Spectrum00578-24-s0001.pdf).** Table S1 to S5.

Open Peer Review

**PEER REVIEW HISTORY (review-history.pdf).** An accounting of the reviewer comments and feedback.

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
