## [Reviewer comments · Microbiology Spectrum]

Microbiology Spectrum

Honey bee gut bacterial strain improved survival and gut microbiota homeostasis in *Apis mellifera* exposed in vivo to clothianidin.

Sarah El Khoury, Jeff Gauthier, Pierre-Luc Mercier, Stephane Moise, Pierre Giovenazzo, and Nicolas Derome

Corresponding Author(s): Nicolas Derome, Universite Laval Department of Biology

Review Timeline:

Submission Date:	March 2, 2024
Editorial Decision:	April 11, 2024
Revision Received:	June 3, 2024
Accepted:	June 4, 2024

Editor: Chengshu Wang

Reviewer(s): Disclosure of reviewer identity is with reference to reviewer comments included in decision letter(s). The following individuals involved in review of your submission have agreed to reveal their identity: Hao Zheng (Reviewer #1); Guan-Hong Wang (Reviewer #2)

Transaction Report:

DOI: <https://doi.org/10.1128/spectrum.00578-24>

Re: Spectrum00578-24 (Honey bee gut bacterial strain improved survival and gut microbiota homeostasis in *Apis mellifera* exposed in vivo to clothianidin.)

Dear Prof. Nicolas Derome:

Thank you for the privilege of reviewing your work. Below you will find my comments, instructions from the Spectrum editorial office, and the reviewer comments.

Revision Guidelines

Sincerely,
Chengshu Wang
Editor
Microbiology Spectrum

Reviewer #1 (Comments for the Author):

This is a revised manuscript previously submitted. I was not one of the reviewers for the last round, but I think there is a major point which may preclude this work for further consideration. The most significant concern is that the two strains used in this work might be an accident isolation from the bee gut. To my knowledge, either *Enterobacter* or *Pantoea* was reported as a native gut commensal for honeybees. It is now quite clear that the honeybee gut is dominated by 8-10 bacterial genera, e.g.,

Gilliamella, Snodgrassella, Bombilactobacillus. I have also never read any reports related to the functions or ecology of Enterobacter or Pantoea in bees. I am afraid if this work is published, it may mislead the knowledge for other scientists in this field. I strongly suggest the authors to at least analyze the abundance of these two strains in the bee gut. Are they prevalent in all samples or only present accidentally in the collected samples from specific locations. I am happy to see the potential probiotic may help bees in pesticide protection, but I feel sorry that the major conclusion and the selection of the probiotic strains are problematic.

Reviewer #2 (Comments for the Author):

In this study, authors selected two strains of bacteria through previous research as potential probiotics that may be used in experiments on host resistance to pesticides. The experimental design and the methods were sufficiently detailed. I have a few comments below:

Line 30 Supplement the economic and ecological value of honey bee and the current threats to them in the importance section.

Line 39 "However....." It should be replaced by such as or for example, which is more consistent with this sentence.

Line 47-49 "little has been investigated so far....." There are many research on the metabolism of pesticides by gut symbionts, and the statements in the article are arbitrary. Try to find some latest research results and cite them in the article.

Line 53 honeybee(s) or honey bee(s)? It should be consistent throughout the text.

Line 105 "et al." This word should be in italics. Please check the full text.

Line 111 In the results section, one or two sentences of summary should be added after describing the results of each part.

Line 135 Honey bee bacteria mainly colonize in the midgut (rectum and ileum), and the dominant populations colonizing different parts of the midgut are different. This should be taken into account when analyzing the dynamics of the microbial interaction network in different gut regions. Refer to these references, Kwong WK, Moran NA (2013) and Martinson VG, Moy J, Moran NA (2012).

Line 165 "gut section" should be replaced by gut region.

Response to Reviewers

Reviewer #1 (Comments for the Author):

This is a revised manuscript previously submitted. I was not one of the reviewers for the last round, but I think there is a major point which may preclude this work for further consideration. The most significant concern is that the two strains used in this work might be an accident isolation from the bee gut. To my knowledge, either *Enterobacter* or *Pantoea* was reported as a native gut commensal for honeybees. It is now quite clear that the honeybee gut is dominated by 8-10 bacterial genera, e.g., *Gilliamella*, *Snodgrassella*, *Bombilactobacillus*. I have also never read any reports related to the functions or ecology of *Enterobacter* or *Pantoea* in bees. I am afraid if this work is published, it may mislead the knowledge for other scientists in this field. I strongly suggest the authors to at least analyze the abundance of these two strains in the bee gut. Are they prevalent in all samples or only present accidentally in the collect samples from specific locations. I am happy to see the potential probiotic may help bees in pesticide protection, but I feel sorry that the major conclusion and the selection of the probiotic strains are problematic.

Author:

We totally agree the gut microbiota of *A. mellifera* workers is commonly dominated by nine bacterial species clusters (Moran et al., 2012), excluding both *Pantoea* and *Enterobacter* from this so-called core microbiome. Yet, the bacterial strains described in our study were isolated from freshly euthanized bee guts directly sampled from a healthy hive environment at our research facility in Deschambault. These strains originate from the environment and are capable of colonizing bee guts, as evidenced by their isolation from healthy live bees. Previous studies have also identified bacteria belonging to the genera *Pantoea* and *Enterobacter* in the gut of bee collected in hives, further validating our findings, and suggesting that these two genera can settle durably in honeybee gut (see seven references below).

Also, Kwong and Moran (2016) demonstrated that individual worker bees can exhibit significant deviations from the typical gut microbiota composition, sometimes showing an increased presence of opportunistic environmental bacteria such as *Pantoea* and *Enterobacter* from the *Enterobacteriaceae* family. Additionally, despite limited understanding of the physiology and ecology of these genera in bees, a previous study (Maruščáková et al., 2020) revealed an intriguing phenomenon. The authors found that the beneficial effect of probiotic (*Lactobacillus sp.*) administration on bees was accompanied by an increase in *Enterobacter sp.* in the bee gut, suggesting a potential indirect, but positive, relationship between the increase in *Enterobacter* and the probiotic effect on bees.

Below is a list of selected references that have isolated or highlighted the presence of both *Pantoea* and *Enterobacter* genera from the bee gut.

- Kwong, W.K. and Moran, N.A. (2016) Gut microbial communities of social bees. *Nat Rev Microbiol* 14, 374–384.
- Piva, S. *et al.* Could honey bees signal the spread of antimicrobial resistance in the environment? *Lett. Appl. Microbiol.* **70**, 349–355 (2020).
- Disayathanoowat, T., Yoshiyama, M., Kimura, K. & Chantawannakul, P. Isolation and characterization of
- bacteria from the midgut of the Asian honey bee (*Apis cerana indica*). *J. Apic. Res.* **51**, 312–319 (2012).
- Gasper, J. *et al.* Enterobacteriaceae in Gut of Honey Bee (*Apis mellifera*) and the Antibiotic Resistance of the Isolates. *spasb* **50**, 69–69 (2017).
- Ganeshprasad, D. N. *et al.* Gut Bacterial Flora of Open Nested Honeybee, *Apis florea*. *Frontiers in Ecology and Evolution* **10**, (2022).
- Rudra Gouda, M. N., Kumaranag, K. M., Ramakrishnan, B. & Subramanian, S. Deciphering the complex interplay between gut microbiota and crop residue breakdown in forager and hive bees (*Apis mellifera* L.). *Current Research in Microbial Sciences* **6**, 100233 (2024).

To further support our findings, below you will find the figures representing the mean of the relative activity of the core microbial ASVs and the *Pantoea* and *Enterobacter* ASVs summarized at the genus level in the different honeybee gut sections (A. midgut, B. ileum and C. rectum) isolated from honeybees) from each experiment conditions. Significant differences between the mean of the relative activity of the different microbial ASVs among the different experimental groups were calculated using pairwise comparisons in a Kruskal Wallis test. Multiple comparisons were assessed using the Dunn test such as “**” <0.05, “***” <0.01.

Our figures support the occurrence of both *Pantoea* and *Enterobacter* genera across almost all experimental conditions.

C. Rectum

Our figures support the occurrence of both *Pantoea* and *Enterobacter* genera across all experimental conditions.

Also, you will find below a table with the ASV activity for each core member and *Pantoea* and *Enterobacter* genera, supporting their active activity across almost all experimental conditions.

Experimental group	Taxa name (spp.)	ASV activity
Sugar control - Midgut	Enterobacter	1101
Sugar control - Midgut	Pantoea	1
Sugar control - Ileum	Enterobacter	7094
Sugar control - Ileum	Pantoea	1
Sugar control - Rectum	Enterobacter	2368
Sugar control - Rectum	Pantoea	-

Pesticide control - Midgut	Enterobacter	-
Pesticide control - Midgut	Pantoea	1
Pesticide control - Ileum	Enterobacter	1225
Pesticide control - Ileum	Pantoea	-
Pesticide control - Rectum	Enterobacter	379
Pesticide control - Rectum	Pantoea	-

Enterobacter control - Midgut	Enterobacter	107
Enterobacter control - Midgut	Pantoea	71
Enterobacter control - Ileum	Enterobacter	139

Enterobacter control - Ileum	Pantoea	41
Enterobacter control - Rectum	Enterobacter	241
Enterobacter control - Rectum	Pantoea	1

Pantoea control - Midgut	Enterobacter	653
Pantoea control - Midgut	Pantoea	-
Pantoea control - Ileum	Enterobacter	456
Pantoea control - Ileum	Pantoea	3
Pantoea control - Rectum	Enterobacter	241
Pantoea control - Rectum	Pantoea	1

Enterobacter + Pesticide- Midgut	Enterobacter	318
Enterobacter + Pesticide- Midgut	Pantoea	13
Enterobacter + Pesticide- Ileum	Enterobacter	812
Enterobacter + Pesticide- Ileum	Pantoea	15
Enterobacter + Pesticide- Rectum	Enterobacter	-
Enterobacter + Pesticide- Rectum	Pantoea	-

Pantoea + Pesticide- Midgut	Enterobacter	526
Pantoea + Pesticide- Midgut	Pantoea	9
Pantoea + Pesticide- Ileum	Enterobacter	1656
Pantoea + Pesticide- Ileum	Pantoea	13
Pantoea + Pesticide- Rectum	Enterobacter	808
Pantoea + Pesticide- Rectum	Pantoea	-

Reviewer #2 (Comments for the Author):

In this study, authors selected two strains of bacteria through previous research as potential probiotics that may be used in experiments on host resistance to pesticides. The experimental design and the methods were sufficiently detailed. I have a few comments below:

Line 30 Supplement the economic and ecological value of honey bee and the current threats to them in the importance section.

Author:

Thank you.

More details were added in the importance section (from lines 34-38):

“Honeybees are essential pollinators that play a critical role in agricultural productivity and ecosystem health. Their economic value is immense, with estimates suggesting that honeybee pollination contributes billions of dollars annually to global agricultural production. This underscores the urgent need to develop effective strategies for safeguarding honeybee populations from environmental stressors.”

Reviewer: Line 39 "However....." It should be replaced by such as or for example, which is more consistent with this sentence.

Author:

Corrected (line 43).

Reviewer: Line 47-49 "little has been investigated so far....." There are many research on the metabolism of pesticides by gut symbionts, and the statements in the article are arbitrary. Try to find some latest research results and cite them in the article.

Author:

We agree with the insightful comment of the reviewer. We added a paragraph from Lines 51-59:

“Notably, these functions, particularly immune response, are partly regulated by the gut microbiota (17–20), which may play a pivotal role in supporting honeybees under pesticide stress. The honeybee genome possesses enzymes for metabolizing xenobiotics but exhibits lower detoxification gene diversity compared to other insects (21), suggesting that honeybees may depend on factors like microbiota for assistance in breaking down harmful molecules. In recent decades, microbes have demonstrated the ability to degrade chemical compounds in natural environments and have been found in various insect orders (22–25), including within the gut microbiota of the honeybee (26).”

Reviewer: Line 53 honeybee(s) or honey bee(s)? It should be consistent throughout the text.

Author:

We agree. We homogenized with “honeybee” in all the manuscript.

Reviewer: Line 105 "et al." This word should be in italics. Please check the full text.

Author:

Thank you. Corrected.

Reviewer Line 111 In the results section, one or two sentences of summary should be added after describing the results of each part.

Author: Thank you for the comment. We have included a summary at the end of each section of the results. Please see below:

Lines 135-138: “In summary, honeybees exposed to clothianidin (0.1 ppb) experienced higher mortality rates compared to the control group. However, supplementation with the *Pantoea* strain significantly improved survival rates throughout the experiment, while supplementation with the *Enterobacter* strain did not show any significant survival benefit over the pesticide control group.”

Lines 154-159: “Taking these results together, clothianidin exposure for 21 days resulted in reduced interactions among gut microbiota in the midgut, ileum, and rectum of honeybees. In the midgut, connectivity decreased significantly, with fewer positive correlations and slightly more negative correlations. In the ileum, interactions between microbial species decreased, while no significant network parameter differences were observed. In the rectum, microbial interactions decreased, but exposed rectum microbiota showed increased connectivity, along with changes in correlation patterns.”

Lines 265-278: “To encapsulate the main outcomes, the 21-day exposure to putative beneficial microbes, including *Enterobacter* and *Pantoea* strains, led to distinct reshaping of honeybee gut microbiota dynamics across various gut sections. Exposure to *Enterobacter* resulted in reduced interactions among ASVs in the midgut and ileum, with increased connectivity and positive correlations observed specifically in the midgut. Conversely, the ileum exhibited decreased interactions and connectivity, accompanied by changes in correlation patterns and decreased neighbourhood connectivity. In the rectum, ASV interactions remained stable but with decreased connectivity and positive correlations, notably impacting non-core members. On the other hand, exposure to *Pantoea* led to decreased interactions and connectivity in the midgut and ileum, accompanied by reduced closeness centrality. Similar trends were observed in the rectum, with decreased ASV interactions, connectivity, and positive correlations, particularly affecting core members like *Bifidobacterium* and *Bartonella*. These findings underscore the differential effects of *Enterobacter* and *Pantoea* strains on gut microbiota composition and network dynamics, suggesting potential implications for gut microbiota homeostasis in honeybees throughout their digestive tract.”

Reviewer Line 135 Honey bee bacteria mainly colonize in the midgut (rectum and ileum), and the dominant populations colonizing different parts of the midgut are different. This

should be taken into account when analyzing the dynamics of the microbial interaction network in different gut regions. Refer to these references, Kwong WK, Moran NA (2013) and Martinson VG, Moy J, Moran NA (2012).

I think there is some confusion here. To our knowledge, rectum and ileum are not considered or defined as midgut subparts. The midgut is composed of proventriculus, ventriculus and malphigian tubules. In our study, the midgut was not separated into these three parts, but investigated as whole gut section, as we did for both ileum and rectum. Knowing that gut sections are documented to have different functions, they provide different resources and therefore different bacterial assemblages (Kwong and Moran. 2016; Khan et al. 2020; Callegari et al. 2021). This is the rationale of the present work to analyse the three most differentiated sections of the honeybee gut.

Reviewer: Line 165 "gut section" should be replaced by gut region.

Author: Corrected (line 189).

Re: Spectrum00578-24R1 (Honey bee gut bacterial strain improved survival and gut microbiota homeostasis in *Apis mellifera* exposed in vivo to clothianidin.)

Dear Prof. Nicolas Derome:

Your manuscript has been accepted, and I am forwarding it to the ASM production staff for publication. Your paper will first be checked to make sure all elements meet the technical requirements. ASM staff will contact you if anything needs to be revised before copyediting and production can begin. Otherwise, you will be notified when your proofs are ready to be viewed.

Sincerely,
Chengshu Wang
Editor
Microbiology Spectrum